# *B. subtilis* Sec and Srp Systems Show Dynamic Adaptations to Different Conditions of Protein Secretion

**DOI:** 10.3390/cells13050377

**Published:** 2024-02-22

**Authors:** Svenja M. Fiedler, Peter L. Graumann

**Affiliations:** Fachbereich Chemie und Zentrum für Synthetische Mikrobiologie, SYNMIKRO, Philipps-Universität Marburg, Hans-Meerwein Straße 4, 35043 Marburg, Germany; svenja.fiedler@chemie.uni-marburg.de

**Keywords:** Gram-positive bacteria, protein secretion, YidC, SecA, SecDF, FtsY, AmyE, *Bacillus subtilis*, single molecule tracking

## Abstract

SecA is a widely conserved ATPase that drives the secretion of proteins across the cell membrane via the SecYEG translocon, while the SRP system is a key player in the insertion of membrane proteins via SecYEG. How SecA gains access to substrate proteins in *Bacillus subtilis* cells and copes with an increase in substrate availability during biotechnologically desired, high-level expression of secreted proteins is poorly understood. Using single molecule tracking, we found that SecA localization closely mimics that of ribosomes, and its molecule dynamics change similarly to those of ribosomes after inhibition of transcription or translation. These data suggest that *B. subtilis* SecA associates with signal peptides as they are synthesized at the ribosome, similar to the SRP system. In agreement with this, SecA is a largely mobile cytosolic protein; only a subset is statically associated with the cell membrane, i.e., likely with the Sec translocon. SecA dynamics were considerably different during the late exponential, transition, and stationary growth phases, revealing that single molecule dynamics considerably alter during different genetic programs in cells. During overproduction of a secretory protein, AmyE, SecA showed the strongest changes during the transition phase, i.e., where general protein secretion is high. To investigate whether the overproduction of AmyE also has an influence on other proteins that interact with SecYEG, we analyzed the dynamics of SecDF, YidC, and FtsY with and without AmyE overproduction. SecDF and YidC did not reveal considerable differences in single molecule dynamics during overexpression, while the SRP component FtsY changed markedly in its behavior and became more statically engaged. These findings indicate that the SRP pathway becomes involved in protein secretion upon an overload of proteins carrying a signal sequence. Thus, our data reveal high plasticity of the SecA and SRP systems in dealing with different needs for protein secretion.

## 1. Introduction

Protein secretion plays an important role for organisms in all kingdoms of life. In bacteria, most of the secreted proteins are translocated via the highly conserved Sec pathway. The main component of the system is the membrane translocase SecYEG, forming an hourglass gate within the membrane. The translocon can be stimulated by membrane protein SecDF, which is bound to different proteins within different organisms, e.g., YajC in *Escherichia coli* or YrbF in *Bacillus subtilis* [1].

Several systems use the SecYEG translocon in order to either transport proteins across or integrate proteins into the cell membrane. Different proteins function as chaperones in each case and find their targets by binding to signal sequences/peptides [2]. One system is the SRP (signal recognition particle) system, which is responsible for integrating proteins into the membrane. The ribonucleoprotein SRP binds to the signal sequence of nascent chains (NCs) at the ribosome and arrests or slows down translation. The complex interacts with the SRP receptor FtsY and diffuses to the translocon. Hand-over of the ribosome NC complex involves a GTP switch between the SRP and FtsY; continued translation drives the insertion of hydrophobic membrane spans into the membrane via lateral opening of the translocon [1,3].

YidC is an integral membrane protein that can integrate proteins into the membrane by itself or, like the SRP pathway, in cooperation with the SecYEG translocon. The YidC pathway is mostly used for smaller proteins, while the SRP/FtsY system is in charge of a large majority of membrane proteins. Like most Gram-positive bacteria, *B. subtilis* has two YidC enzymes (YidC1 and YidC2). While YidC1 is expressed through the whole cell life-cycle, YidC2 is only expressed under certain aspects, e.g., during sporulation. YidCs are redundant, but deleting both proteins is lethal for the cell [4,5].

Secreted proteins are generally shuttled via SecA. SecA is an ATPase that binds to substrate proteins, delivers them post-translationally to the SecYEG translocon, and provides energy for translocation. SecA functions as a dimer but can also be partially functional as a monomer [6,7]. The levels of SecA varies vastly according to different sources, from 46 to 5000 molecules [8,9,10]. In *E. coli*, SecB is involved in the SecA pathway. SecB binds to unfolded proteins at the ribosome and acts as a chaperone [11,12,13]. It has been proposed to hand over secretory proteins to SecA for translocation. On the other hand, SecB has also been implicated in the insertion of integral membrane proteins (IMPs), as it binds to nascent chains of IMPs, besides those of proteins to be secreted [14]. SecA is also suggested to bind to the ribosome directly, so secretion in *E. coli* can clearly be achieved in a SecB-independent pathway. Indeed, SecB is not present in Gram-positive bacteria, and *E. coli secB* is not essential as opposed to *secA*. SecA has been localized partly within the cytosol and largely at the cell membrane. Single molecule analysis showed that the protein can be most likely found in three states: diffusing along the membrane, bound to SecYEG-translocon, and bound to a large, basically immobile complex containing possibly SecYEG and ribosomes translating a protein to be secreted. Using super-resolution microscopy, only a very small amount of proteins could be detected within the cytosol [15], indicating that SecA is largely membrane-associated.

In our study, we aimed to investigate the spatiotemporal behavior of *B. subtilis* SecA compared to that of other proteins involved in protein secretion or membrane integration, FtsY, SecDF, and YidC. To better understand their role in the secretion system, we generated functional fluorescent protein fusions and tested their single molecule dynamics. To further investigate the characteristics of the system, we combined the fusion strains with an overexpression vector that expresses amylase E at a very high level and analyzed changes in protein mobilities between normal and overproduction conditions.

## 2. Methods

### 2.1. Bacterial Strains and Plasmids

All strains used (Appendix A) are based on *Bacillus subtilis* BG214. The fusion strains were constructed via plasmid pSG1164 and used for single crossover integration at the original gene locus. The plasmid contains 500 base pairs of the C-terminal end of the genes for corresponding proteins (without stop codon) in combination with a linker and *mNeongreen* (mNG) [16]. As a resistant cassette chloramphenicol acetyltransferase was used, all strains containing mNG-fusion were grown in a medium containing 5 µg/mL chloramphenicol. After single cross-over integration, the downstream genes were controlled by the xylose promotor (0.1% xylose was added) in case of *secA* and *yidC* constructs; *ftsY* and *secDF* are the last genes in their operon. Xylose induction did not lead to growth defects, which would be expected for the *secA* downstream gene *prfB* encoding for essential protein release factor 2. The *yidC* downstream gene *jag* is a putative RNA binding protein of unknown function; its deletion does not show a detectable loss of fitness (Subtiwiki, Göttingen), so we would not expect an effect of altered Jag levels on the single molecule dynamics of YidC.

For AmyE overproduction, plasmid pM11k_amyEBs of the B.R.A.I.N. AG (Zwingenberg, Germany) was used. The plasmid is self-replicating and contains a HpaII promotor [17] for the constitutive transcription and a high copy number pUB110-like replicon. The plasmid contains a kanamycin-resistance cassette, and therefore, all strains containing the plasmid were grown with 25 µg/mL kanamycin. The plasmid is available upon reasonable request after signing a Material Transfer Agreement with B.R.A.I.N. AG (Zwingenberg, Germany).

### 2.2. Growth Conditions and Growth Curves

All strains were grown in LB medium at 30 °C and 20 rpm. The antibiotics were added to the medium if the strains contained fusions and/or the plasmid pM11k_amyEBs. An overnight culture was used to inoculate a culture to an OD of 0.1. The culture was grown to an OD of 2, 4, or 6. Afterward, the cultures were used for the following experiments. For growth curves, the cultures were inoculated at OD 0.1, and 150 µL was filled in a 96-well plate. Every culture was transferred into 16 wells to achieve a technical multiplicate. The plate was measured and shaken at 30 °C in a plate reader (Victor^2^, PerkinElmer’s, Waltham, MA, USA). The measurements were performed every 10 min for about 24 h. The mean of the measurements of all 16 wells and three days were taken for the growth curve.

### 2.3. Microscopy

A total of 5 µL of the culture was placed onto a clean glass coverslip and topped with an agar pad containing the growth medium. For the agar pad, two smaller slides were used. A total of 100 µL water with 1% agar was placed between the two slides, and after cooling, one of the two slides was removed, and the agar was placed on the sample. Single molecule tracking of the cultures was performed with a customized slim-field setup. The microscope was a Nikon Eclipse Ti-E (Nikon Instruments Inc., Melville, NY, USA), an inverted fluorescence microscope. The camera used was an ImagEM X2 EM-CCD (Hamamatsu Photonics KK, Shizuoka, Japan). The laser of the setup was the center of a 514 nm laser diode (max power 100 mW, TOPTICA BEAM Smart, Pittsfield, MA, USA). The intensity was 20% (about 160 W cm^−2^ in the image plane). A CFI Apochromat objective (TIRF 100 x Oil, NA 1.49) was used. The videos of the samples have 3000 frames. The movies of mNeongreen fusions were cut after 1000 frames, the mVenus fusions were cut after 500 frames, in order to reach single molecule level. After cutting the movies, the cell meshes were set with Oufti [18], and the trajectories’ analysis was performed with Utrack [19]. The final analysis was performed by SMTracker [20,21]. General data for movies see Appendix A.

### 2.4. Phadebas Assay

All cultures were inoculated from the plate and were grown overnight (20 h). The culture contained 2 mL in LB and was incubated at 30 °C. A total of 500 µL of the culture was taken and centrifuged. The supernatant was used for the assay. One Phadebas tablet (Phadebas AB, Uppsala, Sweden) was mixed with 20 mL Phadebas buffer (0.1 M acetic acid, 0.1 M potassium acetate, 5 mM calcium chloride, pH 5). A total of 180 µL was prewarmed, mixed with 20 µL supernatant, and incubated for 10 min. A total of 100 mL of the solution was pipetted into a 96-well plate (no insoluble parts should be in the sample). Each sample was filled in two wells. The measurement was performed in a plate reader with 620 nm via a microplate reader (Tecan Infinite 200 PRO, Tecan, Switzerland). Each well was measured twice. All results of the two wells were taken together, and the means were calculated. The measurement was corrected for normalized cell density (OD_600_) and dilution.

### 2.5. Western Blot

The culture size of the samples for the Western blot was 50 mL. The cultures were centrifuged at 4000 rpm, and the cell pellet was resuspended in 1 mL Western blot buffer (50 mM EDTA, 100 mM NaCl, pH 7.5). A protease inhibitor and 3 mg/mL lysozyme were added. At 37 °C, the cells were lysed. Afterward, the lysed cells were centrifuged, and the lysate was mixed with SDS buffer. The samples were incubated at 60 °C for 1 h. All samples were loaded on a SDS-Gel. The blotting was performed with 8 V for 55 min. Blocking was performed with PBS and 5% milk. The first antibody was an anti-mNeonGreen-Tag antibody (#55074, Cell Signaling Technology, Danvers, MA, USA) and was incubated overnight in PBST with 5% milk. After washing with PBST, the second antibody called peroxidase-conjugated goat-anti-Rabbit-IgG (1:100,000, Sigma Aldrich, St. Louis, MO, USA) was incubated for 1 h (PBST with 5% milk). For the detection of the Western blot, a HR-substrate (Bio-Rad Laboratories, Inc., Hercules, CA, USA) was used.

## 3. Results

### 3.1. Seca Single Molecule Dynamics Alter during Transition into Stationary Phase

SecA is an essential protein and is responsible for the secretion of a vast number of proteins [22]. The protein has been studied in detail in vitro; however, its molecular dynamics in live cells are not well understood. We generated a fusion of SecA with mNeongreen (mNG), which was integrated into the original gene locus, such that SecA-mNG was expressed as the sole source of the protein (Appendix A), under control of the original promoter. Cells carrying the fusion grew indistinguishable from wild type cells (Appendix A), showing that the fusion can functionally replace wildtype SecA. Single molecule dynamics of SecA were studied using single molecule tracking (SMT) fluorescence microscopy, as described before [21,23]. First, we analyzed cells grown to a late log phase, characterized by an optical density (OD) of two (Appendix A). Probabilities for step distances obtained by squared displacement analyses (SQD) could not be well-described using a two-population model; the combined fit for two populations did not cover the obtained step size distribution (Appendix A). However, the distribution was well explained using three Rayleigh fits representing three populations of distinct diffusion constants (Appendix A, Figure 1A). These data suggest that similar to SecA of *E. coli, B. subtilis* SecA exists in three distinct diffusive states. Diffusion is affected by the temperature and pressure, molecule volume, and the forces between molecules. Different diffusion constants of SecA populations thus indicate that the protein co-migrates with other proteins that greatly increase the volume of SecA or that are anchored within the cell, because the other parameters are the same for all molecules within the *B. subtilis* cell. A diffusion constant (D) of 0.75 ± 0.002 µm^2^ s^−1^ can most easily be explained by SecA showing free diffusion [21,23]; according to this idea, 25% of SecA-mNG molecules are freely diffusing (Figure 1B, size of the bubbles represents the population size). A total of 38% of SecA molecules showed an intermediate (medium) diffusion constant of 0.20 ± 0.001 µm^2^ s^−1^, i.e., representing SecA bound to a large but mobile target molecule. In a liquid environment, D scales with the radius of the particle. Because the medium-mobile population had a D more than three-fold lower than that of the freely diffusive population, it is unlikely that this population represents a SecA dimer bound to a single polypeptide (i.e., protein to be secreted), but to a much larger molecule, such as the ribosome. As will become apparent below, SecA localization and dynamics were very similar to those of ribosomes; therefore, we propose that the medium-mobile fraction represents ribosome-bound SecA. The third population of 37% SecA molecules showed very low mobility (0.04 ± 0 µm^2^ s^−1^). SecA is known to be stably engaged with the SecYEG translocon during translocation of secretory proteins [1], therefore we propose that low-mobile/statically positioned SecA represents SecA molecules actively translocating cargo through the membrane-integral translocon.

Interestingly, SecA molecules were markedly enriched close to the cell membrane. Projection of all obtained trajectories into medium-size cells of 3 × 1 µm size showed that SecA molecules were depleted from the center of cells, in spite of the fact that *B. subtilis* cells having the shape of a tube contain most freely diffusing molecules towards the cell center. Membrane enrichment can be explained by an interaction of about one-third of SecA molecules with the membrane-bound SecYEG translocon. This is evidenced by the low mobility of one population of SecA (Figure 1A). However, if the other two-thirds of molecules were freely diffusing, we would not expect depletion of molecules from the middle axis of cells. A similar depletion at sites containing the nucleoids was observed for ribosomes [24]; 70S ribosomes accumulate at subcellular sites surrounding the nucleoids (Appendix A). It is, therefore, possible that the medium-mobile population is largely moving close to the cell membrane because of their affinity to ribosomes. Taken together, our data suggest that 75% of SecA molecules are actively involved in secretion by either interacting with the target molecule at the ribosome (medium-mobile population) or by interacting with the Sec translocon while a protein is secreted through the membrane (static population). 

To further analyze the idea that the static fraction corresponds to molecules interacting with the Sec translocon rather than with ribosomes, we generated confinement maps, in which molecules are sorted into freely diffusing molecules (having any diffusion constant), and molecules that remain confined within a radius corresponding three-times the localization error of this study. The latter population will largely overlap with, but is not identical to, the population of the slow-mobile/static motion. Figure 2 shows that free diffusion of SecA occurs throughout the cytosol, different from membrane proteins SecDF and YidC, where tracks are depleted from the cytosol; tracks apparently moving through the cytosol correspond to rare tracks in the focal plane on the top of the cells, which are occasionally captured, although they are out of focus. Confined tracks of SecA were associated with the cell poles, containing translating 70S ribosomes, and the cell middle (future division sites in large cells where ribosomes accumulate between separated nucleoids), as well as the periphery of the cell, where the Sec translocon is localized. Thus, the static fraction of SecA appears to contain SecA bound to the SecYEG translocon as well as SecA bound to statically positioned ribosomes (e.g., polysomes).

Earlier studies showed that secretion is mostly taking place in the later growth phases [25]. In order to investigate possible changes in the behavior of SecA during different growth phases, we analyzed SecA within three time points of the growth curve, late-exponential, as shown above, transition into the stationary phase (around OD_600_ of 4), and the stationary phase (around OD_600_ of 6). Comparing later time points of growth with the exponential phase, it can be seen that the static population decreased from 37% to 17% during the transition phase, and the medium-mobile fraction increased from 38 to 47% (Figure 1B), while the fast-mobile fraction also increased. These data suggest that less SecA is engaged with the translocon, although in transition to the stationary phase, the amount of proteins secreted into the medium is higher than that in the exponential phase [25], and for that, more active SecA would have been expected. The data also suggest that more SecA might be associated with ribosomes, which is supported by the heat map showing a pattern of localization (Figure 1C) very closely matching that of ribosomes (Appendix A) [24,26,27,28]. In a former study, using a similar experimental setup, qualitatively similar populations for SecA during the transition phase were observed, but with slightly different fraction sizes [24], which differed by roughly 10%. Therefore, we take differences of 10% or less as representing slight experimental differences and biological noise.

During the stationary phase, the population of the fast-mobile molecules further increased and now contained more than half of the molecules (55%), while the static population (19%) is comparable to that during the transition phase (17%), while the medium-mobile population decreased to 27%. These findings suggest that general secretion across the cell membrane is much reduced in the stationary phase, in parallel with strongly decreased general protein synthesis. However, SecA still retained its preferred localization at polar regions, similar to that of ribosomes, with the distinction of a loss of accumulation at the cell middle that is seen during the transition phase (Figure 1C). During growth, ribosomes (and SecA) accumulate before cell division, in between separated nucleoids. Lack of division in the stationary phase abolished central accumulation.

### 3.2. The Localization Pattern of Seca Mimics That of Ribosomes

In *E. coli*, several proteins are functioning as chaperones for newly synthesized proteins containing a signal peptide. The two major molecules fulfilling the chaperone task are SecB and the SRP (signal recognition particle). A protein comparable to SecB has not yet been identified in *B. subtilis* [1]. As explained above, SMT data show that SecA has a comparable pattern to that of ribosomes (Appendix A) [24]. In order to investigate a possible interaction of SecA with the ribosomal complex, we inhibited either transcription or translation, with the help of two antibiotics, rifampicin (Rif) or chloramphenicol (Cm). Both antibiotics target the protein biosynthesis by either reducing the amount of produced mRNA (Rif) or by stopping the peptidyl transferase (Cm) while it is synthesizing proteins and therefore blocking the ribosomal complex on its substrate [29,30]. Figure 3A shows the analysis of the SMT data with and without treatment. Here, the diffusion constants were determined as 0.83 ± 0.001 µm^2^ s^−1^ for the fastest population, 0.18 ± 0 µm^2^ s^−1^ for the medium-mobile population, and 0.04 ± 0 µm^2^ s^−1^ for the static population. The reason for the differences between the diffusion constants here and those in Figure 1 is due to a simultaneous fitting procedure between the different conditions, where a common diffusion constant is determined by SMTracker [20], to better compare changes in molecule dynamics. Variations in single molecule dynamics are thus reflected by changes in population sizes only and not in both, sizes and diffusion constants.

Inhibition of translation by Cm did not change SecA dynamics in a dramatic manner, only the fast-mobile fraction increased slightly (Figure 3A). Similarly, stalling of ribosomes bound to mRNA by Cm treatment did not lead to changes in ribosome dynamics [24]. SecA bound to stalled ribosomes will not be released together with fully translated proteins to be secreted, thus “freezing” SecA dynamics. Conversely, reduction of transcription (a low concentration of Rif was used to avoid rapid cell death observed during complete inhibition of RNA polymerase) led to a marked reduction of the static population, almost by half, while the freely diffusive/fast-mobile fraction strongly gained in size (33% untreated to 47% in the treated condition). Likewise, ribosomes become considerably more dynamic when their substrate, mRNA, is depleted [24]. Reduction of mRNA synthesis leaves ribosomes to finish ongoing rounds of translation, whereby SecA can finish secretion of synthesized proteins but cannot commence new rounds, thus becoming considerably more freely diffusive. 

Importantly, polar enrichment and depletion from the mid-cell plane of SecA remained in place by Cm treatment, but was reduced after the addition of Rif (Figure 3B), similar to the change in the localization of ribosomes (Appendix A). These findings show that SecA behaves very similarly to ribosomes after the addition of different inhibitors. These experiments suggest that SecA is strongly associated with translating ribosomes at polar and mid-cell regions, suggesting that *B. subtilis* SecA interacts with signal peptides as they are released from the ribosomal exit tunnel, in agreement with findings that *E. coli* SecA forms a 1:1 complex with ribosomes in vivo [31].

### 3.3. Seca Single Molecule Dynamics Are Affected by the Overproduction of a Secreted Protein

In all three growth phases investigated above, we observed freely diffusive, and thus unbound, SecA molecules. This would ensure a pool of SecA molecules that can shuttle between Sec translocons and ribosomes. We have previously shown that an increase in the secretory load of SecA (during the transition phase) leads to changes in single molecule dynamics but does not deplete the pool of freely diffusing molecules [23]. In order to test if this holds true for different growth phases, we employed a plasmid for the overproduction of AmyE, kindly provided by the B.R.A.I.N. AG [17]. AmyE is an alpha-amylase that degrades starch in the medium for the supply of carbon sources and energy. AmyE is a protein secreted through the Sec system and released into the medium [25,32]. Appendix A shows that the overproduction of AmyE can be measured in the medium, which is evidence for the correct secretion and the function of the protein. Cells overproducing AmyE grew with a doubling time indistinguishable from that of non-producing cells (Appendix A), indicating that overproduction did not result in considerable stress to the cells. Most of the secretion starts with the transition into the stationary phase [25,33]. In order to see possible changes in behavior during different growth phases again, SecA was tested with the three OD values (Figure 1). Comparing the two conditions in the late-exponential phase (OD 2), the population sizes are comparable, which means that the overproduction of AmyE does not have an effect on SecA at that time (Figure 1B). Note that while AmyE is being produced at this time point, secretion is very low [23]. By looking at the heatmaps in Figure 1C, it can be seen that SecA is slightly more polar in the overproduction strain. During the transition phase, the secretion of AmyE strongly increased [23]. In this phase, SecA becomes more static, the fraction size increased from 17 to 22.5% (Figure 1B), and the heatmaps in Figure 1C reveal that SecA became more polar. In the stationary phase, polar localization became less pronounced, while the static population was also larger (24%) in the cells overproducing AmyE than in wildtype cells (19%). Interestingly, a large difference was observed for the medium-mobile population, which increased from 27% to 44% under overproduction conditions, while the population of freely diffusive molecules dropped from 55% to 32%. Secretion of AmyE was somewhat lower in the stationary phase compared with the transition phase, as shown before [23]. Thus, different changes in population sizes occurred between the transition phase and stationary phase, which are possibly adaptations to different physiological states of the cell. However, under no condition did we observe a drastic loss in freely diffusive SecA molecules, indicating that the SecA system can easily cope with an increased load of proteins containing a signal peptide.

Because of the clear effect of AmyE overproduction on SecA dynamics, we reasoned that this system might be a good tool to prove or disprove involvement of other proteins in general protein secretion. Therefore, we moved the AmyE-overexpression plasmid into strains carrying fluorescent protein fusions to (i) SecDF, thought to cooperate with the SecYEG translocon during protein secretion, to (ii) YidC, involved in membrane insertion of proteins, and (iii) FtsY, the receptor of the signal recognition particle. We also speculated that competition for binding sites, e.g., for SecA or FtsY on the translocon, might be detectable by leading to a reduction of a static, e.g., the SecYEG-bound fraction of FtsY.

### 3.4. Secdf Does Not Show Changes in Single Molecule Dynamics during the Overproduction of Amye

SecDF is stimulating the secretion process via the Sec pathway and is known to be bound to SecYEG [34,35]. A SecDF-mNG fusion was shown to be functional by Strach et al. [23]. The step size distribution can be explained using three Rayleigh fits (Figure 4A), suggesting a freely diffusive population and two distinct fractions in which SecDF is bound to larger complexes. Of note, this could be an overinterpretation of the data, but for the purpose of this study, it is not relevant if two or three populations truly exist. In Figure 4B, the population distribution of SecDF-mNG in the wildtype and the overexpression background of AmyE are shown. The diffusion constants in this analysis are 0.02 ± 0 µm^2^ s^−1^ for the static population, 0.08 ± 0.001 µm^2^ s^−1^ for the slow-mobile population, and 0.60 ± 0.004 µm^2^ s^−1^ for the fastest population. The smallest population, with 14.1%, is the fast-mobile population, likely with freely diffusive molecules searching for their target(s) in the membrane. The slow-mobile and the static population have almost the same size with 44% or 42%, respectively.

If, indeed, two such distinct populations existed in vivo, we would like to offer the following interpretation: the slow-mobile fraction could include proteins that are bound to SecYEG, waiting for a protein that has to be secreted. The static fraction could be bound to SecYEG and a SecA dimer actively secreting a protein. Importantly, the population sizes of SecDF-mNG with and without overexpression of AmyE did not show any striking difference. The static fraction increased by about 10% (considering the entire fraction), which we consider being within biological noise. Thus, increased activity of SecA, as judged from its increased static population (Figure 1B), is not reflected by strong alterations in SecDF dynamics. SecDF heat maps are in agreement with the protein moving within the cell membrane only (Figure 4C).

### 3.5. Ftsy Molecule Dynamics Are Affected by the Overproduction of a Secreted Protein

FtsY is the receptor protein for the SRP (signal recognition particle) and therefore is involved in the insertion of integral membrane proteins. The SRP can recognize its dedicated signal peptides and binds to these at the ribosome, which is shuttled to the Sec translocon together with FtsY [35]. The SRP pathway competes with that of SecA in binding to SecYEG [1]. In our studies, we investigated the influence of the overexpression of AmyE on both pathways. A FtsY-mNG fusion was not functional, and therefore, we generated a mVenus (mV) fusion at the original locus under the original promotor. Cells carrying a *ftsY* deletion grow extremely poorly [3], while the FtsY-mVenus fusion strain showed wildtype-like growth (Appendix A). SMT analysis was performed in the transition phase, where SecA showed the strongest changes in single molecule dynamics.

In Figure 5, the results of the single molecule analysis of the FtsY-mV strain with and without overproduction of AmyE are shown with a three-population fit as the best fit (Figure 5A). The diffusion constants were determined as 0.88 ± 0.004 µm^2^ s^−1^ for the fast-mobile population, 0.12 ± 0.001 µm^2^ s^−1^ for the slow-mobile population, and 0.03 ± 0 µm^2^ s^−1^ for the static population. As suggested for FtsY from *Shewanella putrefaciens* [36], the static population likely reflects the FtsY molecules engaged in the ribosome–NC complex handover at the Sec translocon, while the medium-mobile population is most likely FtsY that is diffusing with a bound SRP/ribosome-NC complex to the membrane, and the fast-mobile population are freely diffusive FtsY molecules. The slow-mobile population comprising 44% of molecules is the biggest population and does not show strong differences between the wildtype and AmyE-overproducing cells (Figure 5B). A total of 36% of the proteins are freely diffusing molecules, indicating that 80% of all FtsY molecules move through the cytosol, either bound to an SRP or searching for a new SRP, and 20% of the molecules are actually membrane-bound. While we cannot exclude that the medium-mobile fraction contains FtsY/SRP/ribosome-NC complexes diffusing along the cell membrane, localization in the heatmaps throughout the cell with a mild preference for cell middle and peripheral sites supports the idea of a large cytosolic fraction of FtsY (Figure 5C), contrary to more pronounced membrane localization found in *E. coli* [37].

Surprisingly, we found a strong shift in population sizes in the overproduction strain (Figure 5B). The static population increased by 40%, from 20% in wildtype cells to 28% in the AmyE strain, while the freely diffusive protein level was reduced from 36 to 22%. The localization of FtsY becomes mildly more polar (Figure 5C). Thus, FtsY appears to become more strongly engaged in static motion and therefore, likely, has enhanced association with the ribosomes and/or the Sec translocon upon overexpression of AmyE. If SecA and FtsY competed for binding to the Sec translocon, we would have expected the opposite effect; thus, our data would suggest that FtsY and the SRP also shuttle newly synthesized AmyE to the translocon. This would agree with the findings showing an overlap between Sec and the SRP systems in the secretion of proteins having highly hydrophobic signal sequences [38], where in case of an increased load of proteins with a signal sequence (e.g., AmyE), the SRP would even take up proteins with a “regular” signal sequence.

### 3.6. Yidc Shows Three Populations Whose Dynamics Are Not Affected by Amye Overproduction

While all proteins mentioned above are directly involved in the Sec system, we also wanted to analyze a protein that can act independently of the Sec translocon. YidC is a membrane protein that translocates proteins into the membrane. It can mediate protein insertion alone or in concert with the Sec translocon. *B. subtilis* has two YidC proteins (YidC1 and YidC2). YidC1 is the major translocase, and the mechanisms of YidC2 are still not completely solved [39,40]. It is thought to take over functions during stress responses. Therefore, we focused on YidC1 and integrated a C-terminal mNG fusion into the original gene locus. In order to test functionality, we performed growth curves of YidC-mNG in a *yidC2* deletion strain in comparison to the wildtype (Appendix A). Cells expressing YidC-mNG in the deletion strain grew indistinguishable from wildtype cells (Appendix A), while a double Δ*yidC1/2* strain was not viable [5,41], showing that the fusion can functionally replace the wildtype protein.

We also used a three-population fit to analyze the SQD data (Figure 6A). About 15% of YidC1-mNG molecules had a diffusion constant of 0.55 ± 0.002 µm^2^ s^−1^ (Figure 6B), most likely representing freely diffusing molecules in the cell membrane. The medium-mobile population (0.08 ± 0 µm^2^ s^−1^) contains more than half of all molecules, possibly single YidC molecules engaged in membrane insertion. The static population (0.02 ± 0 µm^2^ s^−1^) comprising a third of the molecules, 33%, might represent those that are actively inserting proteins into the membrane in connection with the Sec system. As for SecDF, we have not performed additional experiments that can exclude the possibility that only two populations might exist or other scenarios that would explain the existence of three populations. The localization of YidC1 is comparable to a membrane protein (Figure 6C). As for SecDF, changes in population sizes between wildtype and AmyE-overproducing cells are in a range of about less than 10%, which we interpret as no significant difference (biological noise). Thus, as observed for SecDF, and unlike FtsY, there were no strong differences in single molecule dynamics for YidC1 in response to the overproduction of AmyE, indicating the absence of involvement of YidC1 in this process.

## 4. Discussion

The molecular mechanisms of the Sec pathway are well understood in *E. coli*, but for *B. subtilis,* and correspondingly most Gram-positive bacteria, many questions still remain unclear. While *E. coli* cells possess SecB, which can pick up proteins carrying a signal peptide for secretion at the ribosome, it is not clear how SecA interacts with signal peptides in Gram-positive bacteria that lack a SecB protein [42]. In our study, we aimed at investigating how SecA moves, at a single molecule level, relative to ribosomes and relative to the membrane-embedded SecYEG translocon. For comparison, we also monitored the spatiotemporal behavior of membrane insertase YidC1, the SRP component FtsY, and accessory membrane protein SecDF, proteins involved in the integration of membrane proteins or in protein secretion, using real time, single molecule resolution and functional fusions to all mentioned proteins. In a second approach, we studied changes in single molecule dynamics of SecA during overproduction of a secretory protein, to answer the question of how the system copes with a higher than normal load of signal peptides and translocation events. We also studied changes in molecule dynamics of YidC1, FtsY, and SecDF during overproduction, in order to address the question of whether there is cooperation or competition of these factors with SecA.

A key observation is our finding that SecA shows a localization pattern similar to that of ribosomes, which is disturbed when mRNA synthesis is reduced and becomes “frozen” when translation is arrested. Similar to ribosomes, SecA did not markedly change in molecule dynamics or localization during chloramphenicol treatment, but became much more freely diffusive and dispersed in the cells in case of transcription inhibition by rifampicin. These data strongly suggest that *B. subtilis* SecA is directly associated with translating ribosomes, as has been suggested by several studies on *E. coli* SecA, where binding to L23 was demonstrated to be important for efficient translocation [31,43]. Our data thus provide visual evidence for a process in which *B. subtilis* SecA directly picks up proteins containing a signal peptide at the ribosome to guide them to the Sec translocon, in agreement with the lack of a SecB ortholog in *B. subtilis* [42]. In *E. coli*, SecA has been shown to be strongly membrane-associated, and only a few percent are freely diffusive [15]. Our analyses show that *B. subtilis* SecA is largely cytosolic, such that only a minor fraction of SecA can be expected to interact with the Sec translocon. This is supported by SQD analyses: we found that—dependent on the growth phase—one sixth to one third of SecA molecules showed extremely slow/static motion, while the remaining majority of molecules showed diffusion rates similar to that of large protein complexes (e.g., SecA bound to translating ribosomes) or of freely diffusing molecules. Thus, it appears that similar to the SRP system, a majority of signal peptide-translating ribosomes are picked up from the cytosol by SecA to diffuse to the membrane, as opposed to the idea that the chaperone systems might pick up ribosomes at the cell membrane to perform 2D diffusion in search of a translocon. While the latter scenario sounds appealing in terms of efficiency, it appears that model bacteria use a combination of predominant 3D diffusion plus 2D diffusion for membrane protein integration and protein secretion.

We further show that single molecule dynamics of SecA vary between different time points of growth. According to Subtiwiki (http://subtiwiki.uni-goettingen.de) (accessed on 1 January 2024), transcription levels of SecA do not vary strongly between the mid-exponential, transition, and stationary phases, in agreement with the *secA* gene being under the control of the SigA sigma factor [44]. However, the static population of SecA, likely molecules actively involved in the interaction with translating ribosomes or the Sec translocon strongly declined during the transition and stationary phases. This contrasts with the idea that the secretion of exoenzymes, like proteases, lipases, and amylases, commences during the transition phase and continues during stationary growth. Conversely, the transport of enzymes modifying the cell wall for cell growth (e.g., autolysins) likely stops because cell growth ceases. Our observation of a strong reduction in statically engaged SecA molecules at the exit of exponential growth suggests that the secretion of proteins involved in aspects of cell wall remodeling and other growth-related aspects during exponential growth exceeds the extent of the transport of proteins that act in the environment for degradation of extracellular polymeric substances during the transition and stationary phases. Our findings indicate that changes in the need for different proteins to become translocated through the SecYEG system via SecA are accompanied by changes in SecA dynamics, suggesting that the system is quite flexible for altering needs dependent on the cell’s physiology. Likewise, SecA responds dynamically to high levels of expression of AmyE that is efficiently secreted during the transition phase. We show that SecA single molecule dynamics do not change during the late exponential phase, when AmyE is synthesized but not yet secreted, but becomes considerably more static during the transition and stationary phases. While SecA appears to be more strongly engaged in ribosome binding and/or translocon association, the freely diffusive pool of SecA becomes smaller but not depleted, indicating that the system has a high capacity to deal with an additional load of signal peptide-carrying proteins.

We wished to contrast changes in molecule dynamics during high-level AmyE secretion seen for SecA with proteins thought to cooperate in protein translocation, such as SecDF, and with proteins thought to be involved in membrane integration of membrane proteins, like FtsY and YidC. As expected, we did not see differences in single molecule dynamics for YidC between wildtype cells and AmyE-overexpressing cells because YidC is a membrane insertase [39], not known to be involved in secretion. Although amylase secretion is less efficient in the absence of SecDF [45], changes observed were also smaller than 10% with regards to the low-mobile/static fraction. However, intriguingly, we found clear differences in FtsY dynamics between wildtype and overexpressing cells, in that FtsY, which is part of the SRP system for the integration of membrane helices, became more engaged in binding to a statically engaged complex. Because FtsY and SecA bind to SecYEG by direct contacts at similar binding sites and are therefore competing for translocon binding, it would have been expected that a stronger engagement of SecA with the translocon might have led to a reduction in FtsY binding, which would have been reflected by a reduction in the slow-mobile/static fraction [46,47]. The fact that we observed the opposite trend indicates that the SRP system becomes (more strongly) involved in binding to secretory signal peptides and hand over ribosomes to the SecA system or, more likely, directly to the SecYEG translocon. Interestingly, the SRP system has been shown to engage in protein secretion for proteins having a highly hydrophobic signal sequence [38]. We speculate that the *B. subtilis* SRP binds to the AmyE signal peptide when the protein is expressed at a very high level, due to its weak affinity to signal sequences based on their hydrophobic central sequence.

Taken together, our study shows stark differences in the spatiotemporal activity of SecA in *E. coli* and *B. subtilis,* underlining the plasticity of the system. While SecA is mostly bound to the membrane in *E. coli,* the activity of SecB may be of importance because of this, but this is not the case in *B. subtilis*. Here, the growth phase and the quantity of proteins that have to be secreted have an influence on the behavior and localization of SecA, further stressing the flexibility of the secretory system in *Bacillus*.

## Figures and Tables

**Figure 1 cells-13-00377-f001:**
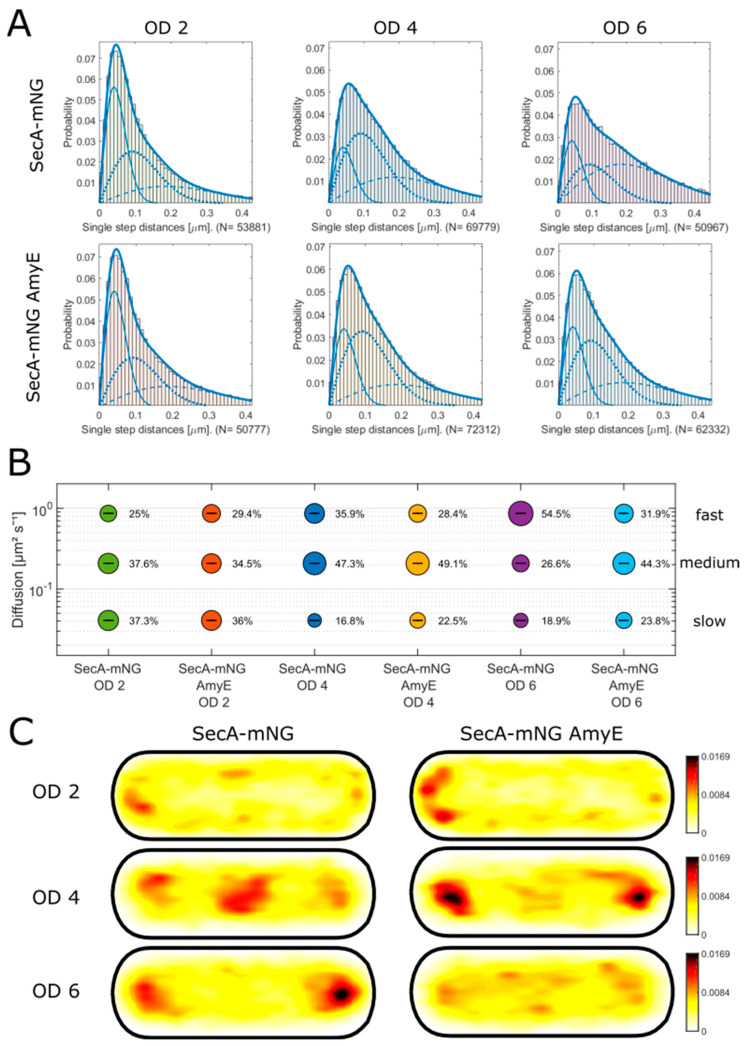
SMT analysis of SecA-mNG during different growth phases. (**A**) Jump distance analysis of the different conditions, using a triple fit indicated by the three lines. The dashed line is the slowest population, the dotted line is the medium-fast population, and the solid line is the fastest population. (**B**) In bubble plots of squared displacement analysis, the bubbles’ size equals the population’s percentage. Simultaneous fitting was employed, leading to the same population diffusion constants. (**C**) Heat maps are presented. They show the localization of the trajectories found in the movies. The conditions named “AmyE” are data from the strain overexpressing AmyE.

**Figure 2 cells-13-00377-f002:**
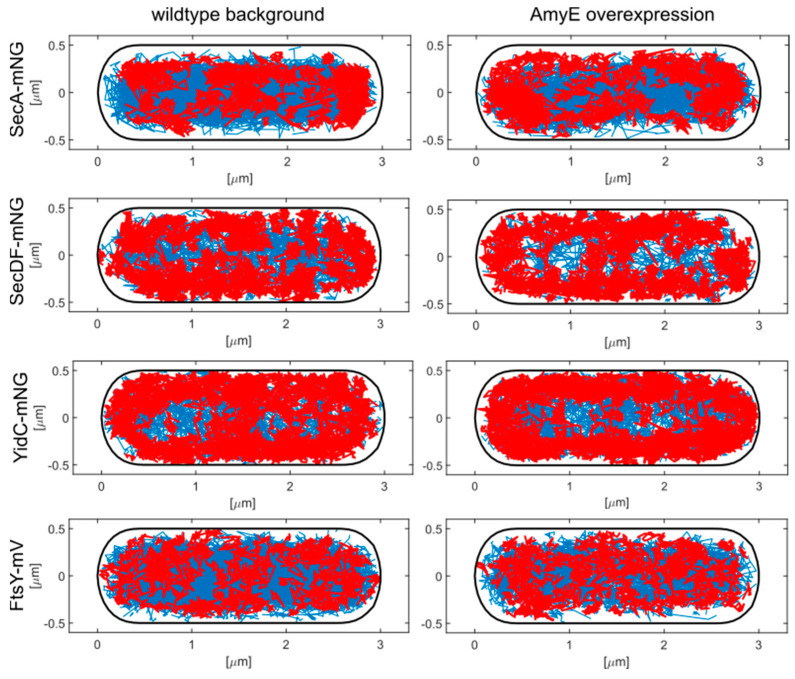
Confinement maps. Tracks were projected into a medium-size cell of 3 × 1 µm (note that *B. subtilis* has an average cell width of about 750 nm). Blue tracks indicate freely diffusive molecules, and red tracks show molecules showing confined motion for at least seven steps within a radius of 120 nm.

**Figure 3 cells-13-00377-f003:**
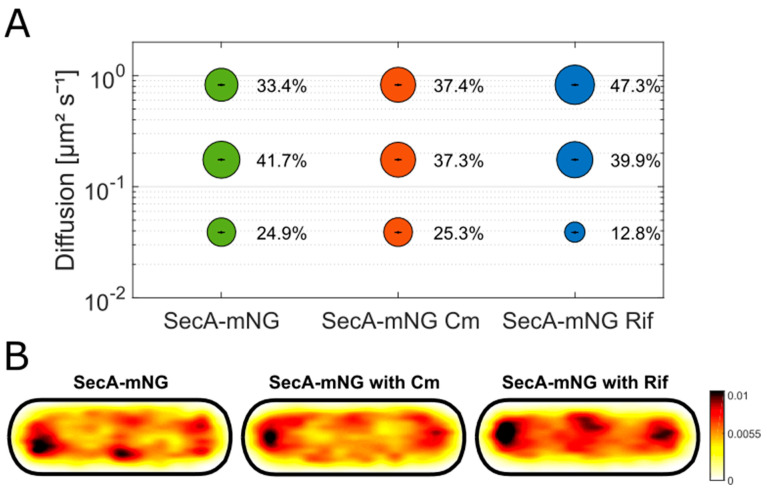
SecA-mNG with and without antibiotic stress. (**A**) shows a bubble plot of the single molecule tracking data of SecA-mNG in the wildtype background (SecA-mNG) and SecA-mNG with chloramphenicol (Cm) and rifampicin (Rif) treatment. (**B**) shows the heatmaps of all trajectories found within the analysis plotted in a uniformed cell (1 µm × 3 µm).

**Figure 4 cells-13-00377-f004:**
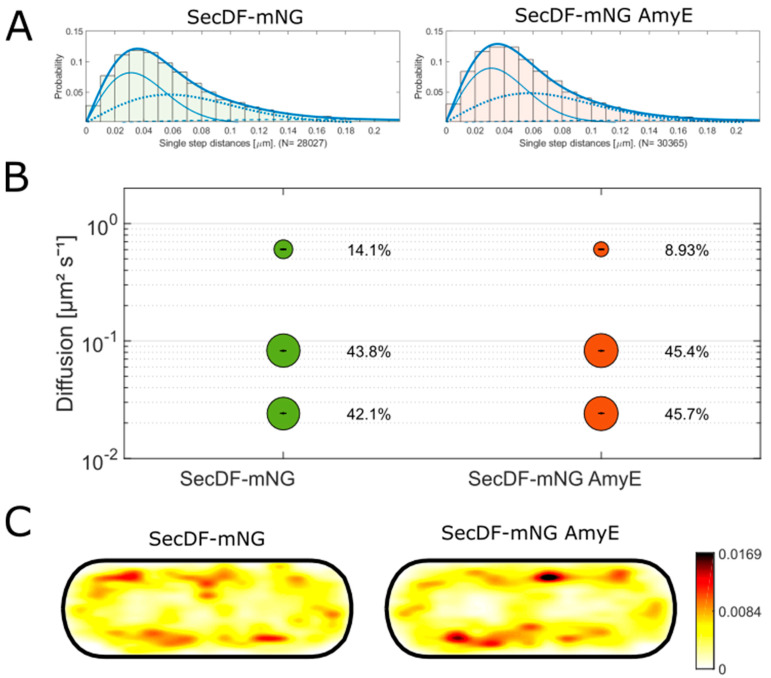
Tracking data of SecDF at OD 4. (**A**) shows the jump distance analysis of SecDF-mNG and SecDF-mNG with AmyE overproduction (SecDF-mNG AmyE). The solid (slow), dashed (middle), and dotted lines represent the populations of the fit. Here, a three-population fit was chosen. (**B**) shows the square displacement analysis (SQD) of the jump distance analysis. The bubbles represent the percentage of the population. (**C**) shows a heatmap of the localization of SecDF-mNG within the cell based on the found trajectories. The cell shown is a uniform cell with a size of 1 µm × 3 µm.

**Figure 5 cells-13-00377-f005:**
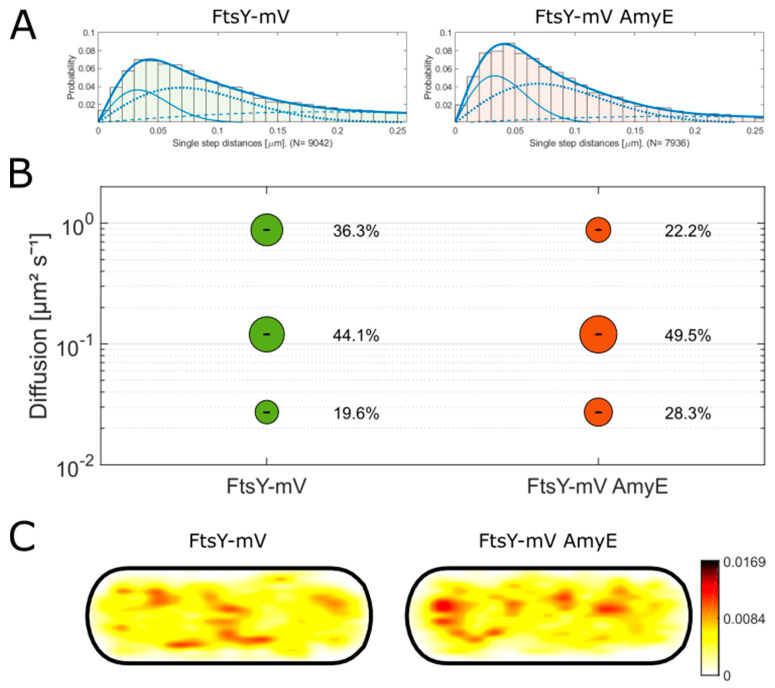
Tracking results of FtsY-mV in a wildtype and AmyE overexpression background. (**A**) Jump distance analysis of FtsY-mV in the wildtype and the overexpression background, a three-population fit was chosen. The solid line represents the most static population. The dashed line is the faster population, and the dotted line is the fastest population. (**B**) SQD (square displacement) analysis of the data in panel A. Bubbles represent the percentage of the population. The diffusion constants were set simultaneously as followed: (**C**) heat maps of localization of FtsY with and without AmyE overproduction.

**Figure 6 cells-13-00377-f006:**
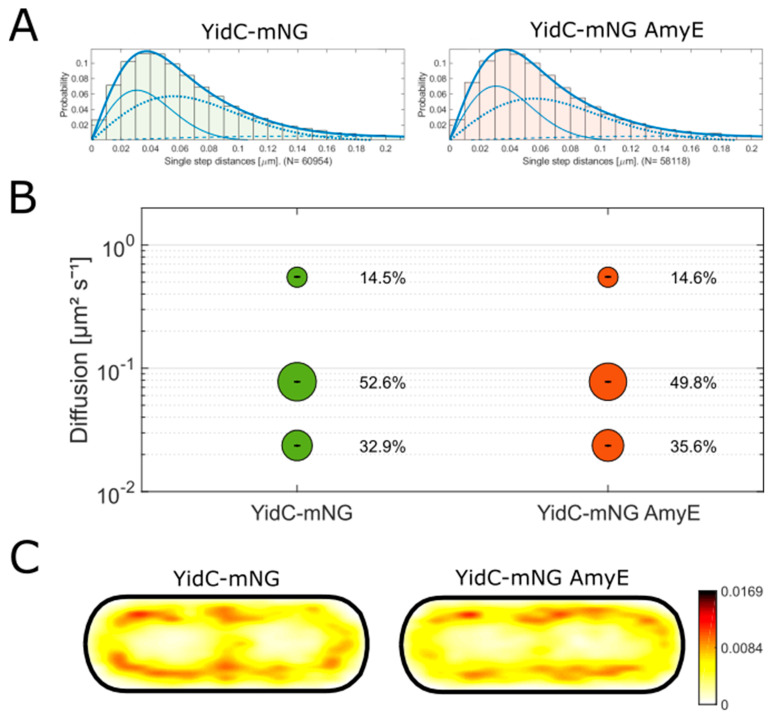
Analysis of SMT data of YidC1-mNG with and without AmyE overproduction. (**A**) Jump distance analysis, the best was determined to be a three-population fit. The solid, dashed, and dotted lines represent the three populations. (**B**) SQD analysis of the data shown in (**A**). A simultaneous fit resulting in the same diffusion constants for both conditions was applied. (**C**) Heat maps of the localization of the tracks of YidC-mNG, projected into a standardized cell of 1 by 3 µm.

## Data Availability

Files for SecA data can be found at DOI: 10.6084/m9.figshare.24881478. Files for data for all other proteins at DOI: 10.6084/m9.figshare.24886374.

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
