# Peer review of "B. subtilis Sec and Srp Systems Show Dynamic Adaptations to Different Conditions of Protein Secretion"

_cells, 2024, doi:10.3390/cells13050377_

Round 1

Reviewer 1 Report

Comments and Suggestions for Authors

This study expands on the authors’ long-term investigations of spatial dynamics of Sec components in model bacteria using real-time single molecule tracking of fluorescently-tagged subunits. Following up on their BMC Biology paper, the authors now interrogate the spatial localization of SecA, FtsY, SecDF, and YidC, without and with overproduction of the secreted substrate AmyE. Most importantly, SecA localization patterns in B. subtilis differ appreciably from those in E. coli. Specifically, localization and molecular dynamics data suggest that in B. subtilis the majority of SecA associates with signal peptides as they are being synthesized at the ribosome, e.g., SecA is predominantly a mobile cytosolic protein as opposed to being statically bound at the membrane through binding  SecYEG translocation.  The authors attribute this to the fact that, unlike E. coli, B. subtilis and other G+ bacteria lack SecB chaperone; apparently SecA fulfills this chaperone function. The authors also showed that SecA exhibits profoundly different molecular dynamics in cells at the late exponential, transition, and stationary growth phases, exhibiting localization patterns consistent with expected reductions in protein secretion during stationary phase. Finally, the authors went on to compare SecA molecular dynamics in the absence and presence of AmyE overproduction  with those of other Sec components, SecDF, FtsY, and YidC. An important and surprising outcome of this line of study is that SecA and FtsY/SRP do not appear to compete for the Sec translocon under conditions of AmyE overproduction, in contrast to expectations that SecA engagement would predominate.  Overall, this study reports new findings that advance our mechanistic understanding of the Sec pathway in a model G+ bacterium, and that distinguish this pathway from the equivalent pathway in a model G- bacterium. The work appears to be rigorously executed, although I am not an expert in single-molecule tracking. The authors offer plausible explanations for their findings, and nicely integrate their results with the broader literature; this greatly helps non-expert readers to follow and interpret the lines of experimentation.  This reviewer identified only a few issues for the authors to address in this well-crafted study:

1.               L. 161. Fig. 1. Fig. S4. The data presented in Fig. 1.  and S4 are highly similar to the first panel of Fig. 9A in the authors BMC Biology paper, although there are also quantitative differences in the 3 predicted cellular states of SecA without and with AmyE overproduction. Perhaps the authors should clearly justify inclusion of these data here with more explicit statements accounting for similarities/differences between the two datasets (either in Results or Discussion section).

2.               L. 306. Fig. 4. Again, the data presented in Fig. 4 resemble those in Fig. 9 of the BMC Biol. Paper. The authors should account for similarities/differences in the two datasets.

3.               While most of the data interpretation is reasonable, as there is no way to test these interpretations,  the reader has no choice but to accept them.  For the most fundamental findings, for example, that SecA is predominantly mobile in B. subtilis whereas it is mostly membrane-bound in E. coli, it would seem straightforward to test this with biochemical/crosslinking, or genetic approaches. This, of course, is for a future study. 

4.               Methodology: This reviewer has concerns over the single-crossover strategy for genes embedded within an operon.  I can’t remember the operon structure of the sec genes in B. subtilis, but in the Methods section it is stated that, after plasmid integration for subunit tagging, the downstream genes are controlled by induction of a downward-reading xylose promoter.  This non-native expression system could easily lead to artefacts in downstream-encoded protein levels, which in turn could significantly affect molecular dynamics studies of subunits under investigation.  Is there a way to evaluate this possibility?

5.               Methodology:  I understand the rationale for evaluating effects of AmyE overproduction on cellular localization and dynamics of Sec components. However, in this artificial system, overproduced AmyE could disrupt membrane integrity or elicit envelope stress responses that alter Sec component molecular dynamics in ways completely unrelated to those that the authors envision. How can the authors exclude the possibility that experimental artefacts arise from artificially overproducing a secreted protein?  

6.               Figures. It would be useful to the reader if the figures contained identifiers for the slow, medium-fast, and fast populations, rather than have these identified only in the legends. 

6.               Overall, English grammar is excellent, but some editing is still needed. Examples include but are not limited to:

A.              L. 48. Rephrase: YidC is another integral membrane protein that integrates proteins into the membrane, but other than the SRP pathway, YidC can use the SecYEG translocon but is not dependent on it.

B.              L. 60. Rephase: In E. coli, SecB has been involved in the SecA pathway. (It is involved).

C.              L. 228. Run-on sentence: However, SecA still retained preferred localization at polar regions, similar to ribosomes, with the distinction of a loss of accumulation at the cell middle seen during the transition phase (Fig. 1C), due to an arrest in cell division, and thus lack of two separated nucleoids before cell division, which results in an accumulation of ribosomes in the cell middle in large, growing cells. 

d.               L. 302.  “observed”? 

7.               L. 248. Add (Chl) to this phrase – “by stopping the peptidyl transferase while it is synthesizing proteins and, therefore, blocking the ribosomal complex on its substrate [28,29].”

8.               L. 284.  This should cite Fig. S5, not Fig. S4. 

Comments on the Quality of English Language

English is excellent; as with all initial submissions, some revisions are needed, but these are very minor. Overall, the manuscript is excellently presented. 

Reviewer 2 Report

Comments and Suggestions for Authors

In this study, the authors demostrated the dynamic behavior of these components of Sec secretion pathway. Generally, the study is interesting and the results were clearly elucidated. However, there also are some questions need to be addressed.

Firstly, the keynote of the study should be more precisely and clearly presented in the abstract. The conclusion should be more highlighted. We know there are several proteins involving into the secretion process. However, in this study, which process or which protein is the key point the authors would like to study? what is the relationship between the behaviors of these proteins, such as SecA, presented by the authors, and the different need for protein secretion by SecA? It is generally regarded that SRP highly related to membrane protein targeting to the inner-membrane. How is the SRP used in the application need mentioned by the authors? 

How does migration rate reflect the movement state of SecA and thereby indicate which physiological process it is involved in?

Why does the migration rate of SecA show no significant change after ribosome inhibition, while it changes more significantly after transcription inhibition? The reasons for this are not explained clearly. How do these results support the conclusion that "SecA's localization pattern is similar to that of ribosomes"?

What is the relationship between the dynamics of SecDF, YidC, and FtsY and the elucidation of SecA's single-molecule dynamics? What scientific questions can these results explain?

The article involves multiple research subjects. What is the key scientific question that the article aims to clarify? The core is not highlighted.

Round 2

Reviewer 2 Report

Comments and Suggestions for Authors

The authors have addressed all my questions, and they also revised the manuscript carefully. So, I recommendate accepting the revised manuscript.